# Pan-Plastome of Greater Yam (*Dioscorea alata*) in China: Intraspecific Genetic Variation, Comparative Genomics, and Phylogenetic Analyses

**DOI:** 10.3390/ijms24043341

**Published:** 2023-02-07

**Authors:** Rui-Sen Lu, Ke Hu, Feng-Jiao Zhang, Xiao-Qin Sun, Min Chen, Yan-Mei Zhang

**Affiliations:** 1Institute of Botany, Jiangsu Province and Chinese Academy of Sciences, Nanjing 210014, China; 2Jiangsu Key Laboratory for the Research and Utilization of Plant Resources, Nanjing 210014, China

**Keywords:** *Dioscorea alata*, pan-plastome, haplotype variations, comparative genomics, DNA barcodes, phylogenetic analyses

## Abstract

*Dioscorea alata* L. (Dioscoreaceae), commonly known as greater yam, water yam, or winged yam, is a popular tuber vegetable/food crop worldwide, with nutritional, health, and economical importance. China is an important domestication center of *D. alata*, and hundreds of cultivars (accessions) have been established. However, genetic variations among Chinese accessions remain ambiguous, and genomic resources currently available for the molecular breeding of this species in China are very scarce. In this study, we generated the first pan-plastome of *D. alata,* based on 44 Chinese accessions and 8 African accessions, and investigated the genetic variations, plastome evolution, and phylogenetic relationships within *D. alata* and among members of the section *Enantiophyllum*. The *D. alata* pan-plastome encoded 113 unique genes and ranged in size from 153,114 to 153,161 bp. A total of four whole-plastome haplotypes (Haps I–IV) were identified in the Chinese accessions, showing no geographical differentiation, while all eight African accessions shared the same whole-plastome haplotype (Hap I). Comparative genomic analyses revealed that all four whole plastome haplotypes harbored identical GC content, gene content, gene order, and IR/SC boundary structures, which were also highly congruent with other species of *Enantiophyllum*. In addition, four highly divergent regions, i.e., *trnC*–*petN*, *trnL*–*rpl32*, *ndhD*–*ccsA*, and exon 3 of *clpP,* were identified as potential DNA barcodes. Phylogenetic analyses clearly separated all the *D. alata* accessions into four distinct clades corresponding to the four haplotypes, and strongly supported that *D. alata* was more closely related to *D. brevipetiolata* and *D. glabra* than *D. cirrhosa*, *D. japonica,* and *D. polystachya*. Overall, these results not only revealed the genetic variations among Chinese *D. alata* accessions, but also provided the necessary groundwork for molecular-assisted breeding and industrial utilization of this species.

## 1. Introduction

*Dioscorea alata* L. (also called ‘greater yam’, ‘water yam’, or ‘winged yam’) is a perennial climbing herb that belongs to the section *Enantiophyllum* of the genus *Dioscorea* L. (Dioscoreaceae) [1]. It is the most widely cultivated and the second most produced yam species (*Dioscorea* spp.) across the world, including Asia, Africa, the Pacific, the Caribbean, and America [2,3]. Recently, this species has gained increasing interest from both consumers and producers, due to several valuable attributes. First, *D. alata* is an important traditional food crop and considered a rich source of carbohydrates (75–84%), proteins (~7.4%), and vitamin C (13.0–24.7 mg/100 g) [4,5]. Second, *D. alata* has sufficient capacity to adapt to diverse agro-ecological and climatic environments and can be easily propagated vegetatively by tuber pieces [6,7,8]. Third, its tubers often have a delicate taste, an attractive tuber appearance, and an excellent long-term (4–6 months) storage ability [9,10]. Last, *D. alata* is rich in secondary metabolite compounds, e.g., phenolic acids, flavonoids, and anthocyanins, and thus has great potential as a healthy and functional food [11]. There is no doubt that this species has diverse application potential to be utilized in different food and non-food-based industries globally.

China is an important domestication center for *D. alata*, where hundreds of cultivars/landraces have been developed over its long cultivation history [12]. However, the generation of these cultivars depends largely on the human selection of somatic mutations [13,14], whereas genetic improvement of *D. alata* lags far behind that of most other crops. The success of genetic improvement programs mainly lies in the level of genetic variation within species, which will guide the selection of suitable parents for breeding hybrids, and the subsequent development of improved cultivars [6,15]. To date, however, genetic variations of Chinese *D. alata* accessions remain ambiguous, although several simple sequence repeat (SSR)-based studies have been conducted. For example, Wu et al. [12] divided Chinese accessions of *D. alata* into two major clades, based on the unweighted pair-group method with arithmetic mean (UPGMA) and principal coordinate analysis (PCA), while Chen et al. [16] revealed the existence of three sub-populations in Chinese accessions based on a model-based population structure analysis. Thus, resolving genetic variations in Chinese *D. alata* accessions is an essential step toward the molecular breeding of this economically significant species in China.

Plastomes in angiosperms typically possess a highly conserved quadripartite circular structure, with sizes ranging from 100 to 200 kb, and encode many key proteins in relation to photosynthesis and other major cellular functions, including the synthesis of starch, fatty acids, pigments, and amino acids [17,18,19]. Due to the absence of recombination, low levels of nucleotide substitution, uniparental mode of inheritance, and small effective population size [20,21], plastome sequences have been widely used for accurate species identification and phylogenetic inference in different plant lineages [21,22,23]. More recently, pan-plastomes have been developed to capture inter- and intraspecific variations [24,25,26]. For example, Magdy et al. [25] constructed high-resolution pan-plastomes from 321 *Capsicum* accessions to differentiate cultivars and lineages. Wang et al. [26] de novo assembled the plastomes of 316 *Nelumbo* accessions to construct a reliable pan-plastome map and investigate the phylogeography and genetic diversity among them. Furthermore, these studies also indicated that pan-plastomes have several advantages over nuclear pan-genomes, such as (i) easier assembly that is not affected by polyploids, (ii) much more complete reference sequences for assembly, annotation, and comparison, and (iii) the absence of large duplicate syntenic regions, thus relieving the effects of gene paralogy in molecular systematics [26].

Here, we de novo assembled plastomes for 44 Chinese accessions and 8 African accessions, and leveraged these pan-plastome-scale resources to examine the genetic variations within the Chinese accessions. Coupling these with five previously published plastomes of the section *Enantiophyllum* (one accession each for *D. brevipetiolata*, *D. cirrhosa*, *D. glabra*, *D. japonica*, and *D. polystachya*, respectively), we also developed reliable molecular markers for further studies regarding the taxonomy, phylogeny, and population genetics of *Enantiophyllum* species, and reconstructed the phylogenetic relationships within this section.

## 2. Results

### 2.1. Plastome Structure and Organization of D. alata

A total of 52 plastomes were successfully de novo assembled and analyzed (Table 1). The plastome size across most of the accessions (34 of them, Haps I and II) was 153,161 bp, while 16 and 2 accessions had reduced plastome sizes of 153,158 (Hap III) and 153,114 bp (Hap IV), respectively (Figure 1, Table 1). All of these plastomes retained the typical quadripartite structure of angiosperm plastomes, which consisted of a pair of inverted repeat (IR) regions (25,464 bp), separated by a large single copy (LSC) region of 83,351–83,415 bp and a small single copy (SSC) region of 18,815–18,836 bp (Figure 1). The GC content in the whole genome sequence (37.0%), and also in the LSC (34.8%), SSC (31.0%), and IR (43.0%) regions, were completely identical among all the *D. alata* accessions.

The plastomes of all 52 *D. alata* accessions encoded identical 113 unique genes, including 79 protein-coding genes (PCGs), 30 transfer RNA (tRNA) genes, and 4 ribosomal RNA (rRNA) genes, 19 of which (7 PCGs, 8 tRNA genes, and all 4 rRNA) were duplicated in the IRs, giving a total of 132 genes (Figure 1, Appendix A). Among the unique genes, eight of the PCGs (i.e., *atpF*, *petB*, *petD*, *ndhA*, *ndhB*, *rpoC1*, *rpl2*, and *rpl16*) and six of the tRNAs (*trnK*-UUU, *trnG*-UCC, *trn*L-UAA, *trn*V-UAC, *trnI*-GAU, and *trnA*-UGC) contained a single intron, while three PCGs (*ycf3*, *rps12*, and *clpP*) possessed two introns (Figure 1, Appendix A). The *rps12* gene consists of three exons that are trans-spliced together: exons 2 and 3 are proximal and located in the IRs, while exon 1 is ~29.0 kb away from the nearest copy of exons 2 and 3 and ~70.5 kb away from its distal repeat copy. An intact gene encoding initiation factor IF1 (*infA*) was present, while the *rps16* gene was independently lost in the *D. alata* plastomes (Figure 1, Appendix A).

### 2.2. Plastome Polymorphisms in D. alata

The alignment of plastomes from the 52 *D. alata* accessions resulted in a data matrix of 153,206 characters, of which 69 were variable. In total, four whole-plastome haplotypes were identified (see details in Appendix A), with the overall values for the haplotype (*H*d) and nucleotide (*π*) diversities were 0.51 and 0.05 × 10^−3^, respectively. Among the four haplotypes (Haps), (i) Hap I was the most prevalent haplotype (found in 63.5% of all the accessions), and present in all the African accessions; (ii) Hap II was only found in one accession from Hainan Province, and just had one single nucleotide polymorphism difference (SNP) from the Hap I; (iii) Hap III occurred in six out of the eight provinces of mainland China, including Fujian (four accessions), Hainan (two accessions), Jiangsu (one accession), Jiangxi (five accessions), Shandong (one accession), and Zhejiang (three accessions) Provinces; and (iv) Hap IV was shared by two accessions, i.e., CDa020 from Hunan Province and CDa044 from Zhejiang Province (Table 1). Tajima’s *D* test in the whole-plastome level of the *D. alata* showed a negative value (–1.80), which may indicate a recent population expansion or some purifying selection in this species.

### 2.3. Whole Plastome Comparison at Intra- and Inter-Specific Levels

The comprehensive comparisons of the nine *Enantiophyllum* plastomes (i.e., four whole plastome haplotypes of *D. alata*, and one representative plastome each for *D. brevipetiolata*, *D. cirrhosa*, *D. glabra*, *D. japonica*, and *D. polystachya*, respectively) revealed that the overall sequence similarity was higher than 99.9% among all the haplotypes of *D. alata*, while some intergenic spacers showed a high degree of divergence (<70% similarity) between *D. alata* and another five species of *Enantiophyllum* (Figure 2). In addition, at both intra- and inter-specific levels, the coding regions were more conserved than the noncoding regions, including intergenic spacers and introns, and the IRs showed less divergence than the LSC and SSC regions (Figure 2).

The IR/SC junctions were identical or nearly identical, not only among the haplotypes of *D. alata*, but also even between *D. alata* and its close relatives (Figure 3). In general, the *rps19* gene extended 62–63 bp into the IRA region at the junction of the LSC/IRA (J_LA_), creating a duplicated pseudogene at the IRB region (pseudogene not shown). The *ycf1* gene crossed the SSC/IRA junction (J_SA_), with the same length (324 bp) in the IRA region, and a length of 5241 bp to 5271 bp in the SSC region. Similarly, the *ndhF* gene was located in the SSC/IRB junctions (J_SB_), with 2231 bp in the SSC region and 7 bp in the IRB region across all the plastomes (Figure 3). In addition, the *trnH*-GUG genes were all located in the IR regions, at a 191 bp distance from their adjacent IR/SC junctions (Figure 3).

### 2.4. Dispersed Repeats and SSRs

A total of 292 dispersed repeats were identified across all nine *Enantiophyllum* plastomes, including four whole-plastome haplotypes of *D. alata*, and one plastome each for *D. brevipetiolata*, *D. cirrhosa*, *D. glabra*, *D. japonica*, and *D. polystachya*, respectively. Among all the identified repeats, the forward and palindromic repeats were considerably higher in number than the reverse and complement repeats (Figure 4A), and the repeat lengths of 30–39 bp were the most common (Figure 4B). Within *D. alata*, Haps I, II, and III shared the same 13 forward repeats and 14 palindromic repeats, while Hap IV harbored 10 forward repeats and 13 palindromic repeats (Figure 4A). At the inter-specific level, *D. japonica* and *D. polystachya* contained the most repeats (16 forward, 14 reverse, 21 palindromic, and 10 complement repeats), followed by *D. alata*, *D. brevipetiolata* (11 forward and 14 palindromic repeats), and *D. glabra* (10 forward, 1 reverse and 11 palindromic repeats), while *D. cirrhosa* (7 forward and 12 palindromic repeats) contained the fewest (Figure 4A).

The total number of SSRs in the *Enantiophyllum* plastomes ranged from 62 (*D. glabra*) to 69 (*D. japonica*), and the types of repeat motifs also varied among these plastomes (Figure 5, Appendix A). Of all the SSRs, the most abundant SSR type was mononucleotides (all A/T repeats), varying from 34 (53.97%) in *D. cirrhosa* to 40 (58.82%) in Haps I and II of *D. alata*, followed by dinucleotides (11 in *D. glabra* to 13 in *D. japonica*), and trinucleotides (7 in *D. glabra* to 10 in *D. japonica* and *D. polystachya*), while tetranucleotides (5 in *D. brevipetiolata* and 4 in the other eight *Enantiophyllum* plastomes), pentanucleotides (3 in *D. alata* and *D. cirrhosa* to 4 in *D. brevipetiolata*, *D. glabra*, *D. japonica*, and *D. polystachya*), and hexanucleotides (1 in each plastome) were rarely observed in the *Enantiophyllum* plastomes (Figure 5, Appendix A). The dominant dinucleotide repeat types were AT/TA, accounting for 12.90% (*D. glabra*)–15.87% (*D. cirrhosa*) of all SSRs in each plastome, while the GA (one in each plastome) and TC (two in each plastome) motifs were least abundant (Figure 5, Appendix A). The abundant A/T and AT/TA repeats likely contributed to the overall AT richness of the *Enantiophyllum* plastomes.

### 2.5. Plastome-Divergent Hotspots in the Enantiophyllum

Based on the whole-plastome alignment of nine *Enantiophyllum* plastomes (i.e., four whole plastome haplotypes of *D. alata*, and one plastome each for *D. brevipetiolata*, *D. cirrhosa*, *D. glabra*, *D. japonica*, and *D. polystachya*, respectively), a total of 120 regions (60 CDS, 45 IGS, 11 introns, and four tRNAs) were subjected to analyses of nucleotide diversity (*π*). The *π* values for these regions ranged from 2.44 × 10^−4^ (IGS *rps12*–*trn*V) to 1.90 × 10^−2^ (IGS *trn*C–*petN*), with an average of 2.90 × 10^−3^ (Figure 6). The non-coding regions (including IGS and introns) showed higher average *π* value (3.83 × 10^−3^) than the tRNA (average *π* = 3.42 × 10^−3^) and CDS (average *π* = 2.00 × 10^−3^) regions, echoing the finding that the non-coding regions were more variable than the coding regions (Figure 2). Four *π* value peaks (*π* > 8.50 × 10^−3^), i.e., *trnC*–*petN*, *trnL*–*rpl32*, *ndhD*–*ccsA*, and exon 3 of *clpP,* were recognized as divergent hotspots and could be used as DNA barcodes at different taxonomic levels within the *Enantiophyllum*.

### 2.6. Phylogenetic Relationships within D. alata and among Members of Enantiophyllum

The topologies of the maximum likelihood (ML) and Bayesian inference (BI) trees based on whole-plastome sequences and 79 shared protein-coding regions were totally identical, with 100% bootstrap (BS) values and 1.0 Bayesian posterior probabilities (PP) at the seven main nodes (Figure 7). All the phylogenetic topologies identically supported the monophyly of *D. alata*, which was sister to the *D. brevipetiolata*–*D. glabra* group. These three species further shared a common ancestor with *D. cirrhosa* and its sister group *D. japonica*–*D. polystachya*. The 52 accessions of the *D. alata* could be further divided into four clades (clades I–V), corresponding to the four haplotypes. Clade I (25 Chinese accessions and all 8 African accessions) and Clade II (CDa017 from Hainan Province) were, together, sister to clade III (16 Chinese accessions), while clade IV (CDa020 from Hunan Province and CDa044 from Zhejiang Province) occupied an early diverging (‘basal’) position within the *D. alata* (Figure 7).

## 3. Discussion

### 3.1. Plastome Evolution in D. alata and Its Closely Related Species

Previous studies have reported that plastomes within a species are highly conserved in terms of genomic structure, GC content, gene content, and the synteny of gene order [26,28], with *D. alata* being no exception in this regard. The pan-plastome of *D. alata* constructed here indicated that all 52 accessions (153,114–153,161 bp) from different geographical areas possessed the typical quadripartite structure of land plant plastomes, with a pair of IR regions (25,464 bp) separating the LSC (83,351–83,415 bp) and SSC (18,815–18,836 bp) regions, and encoded the same 113 unique genes, including 79 PCGs, 30 tRNAs, and 4 rRNAs (Figure 1, Appendix A). All these sequenced plastomes also shared the same overall GC content (37.0%), higher than that in the LSC (34.8%) and SSC (31.0%) regions, but lower than that in the IR regions (43.0%), likely due to the high GC content (55.3%) of the four rRNAs. In addition, no gene arrangements were detected either within *D. alata* or in comparison to its closely related species (i.e., *D. brevipetiolata*, *D. cirrhosa*, *D. glabra*, *D. japonica,* and *D. polystachya*) (Figure 2). These findings were also largely consistent with the characteristics of previously published plastomes of *Dioscorea*, which showed that the plastomes in this genus were well-conserved [29,30,31].

Lineage-specific expansions and contractions of IR/SC boundaries, which are common in angiosperms, often bring about the gain or loss of a small number of genes and length variations of the plastomes [32,33,34]. In this study, the locations of the IR/SC boundaries were totally identical, not only across the four whole-plastome haplotypes of *D. alata*, but also between *D. alata* and its four closely related species, i.e., *D. brevipetiolata*, *D. cirrhosa*, *D. glabra*, and *D. japonica* (Figure 3). Given that the difference in the IRA/SSC boundary between *D. polystachya* and the other plastomes is very trivial (Figure 3), these results provided a further proof of the conserved nature of the *Enantiophyllum* plastomes. In addition, in contrast to most monocots, with their IRs expanding and encompassing the *rps19*-*trn*H gene cluster (e.g., Arecales, Dasypogonaceae, Poales, Zingiberales, Asparagales, and Commelinales) [28,32], all the LSC/IRA boundaries of the *Enantiophyllum* plastomes were located within the *rps19* gene (Figure 3).

### 3.2. Plastome-Derived Markers for Species/Cultivar Delimitation in Enantiophyllum

Although DNA barcoding has been introduced to the taxonomy of *Dioscorea* and has expedited species identification and relationships in the different sections and/or major clades of this genus, the loci commonly used, i.e., plastid genes (e.g., *atpB*, *matK*, *rbcL*), supplemented with intermediately variable IGS regions (e.g., *trn*L–*trn*F and *psbA*–*trn*H) and/or internal transcribed spacers (ITS) of nuclear ribosomal DNA genes, always showed low discriminatory power at low taxonomic levels, including within the *Enantiophyllum* [35,36,37,38,39]. Accordingly, additional hypervariable regions should be developed for the taxonomic and phylogenetic analysis of a specific section/major clade of *Dioscorea*. In this study, the nucleotide diversity analysis of 120 regions (60 CDS, 45 IGS, 11 introns, and four tRNA regions) extracted from the nine *Enantiophyllum* plastomes (i.e., four whole-plastome haplotypes of *D. alata*, and one plastome each for *D. brevipetiolata*, *D. cirrhosa*, *D. glabra*, *D. japonica*, and *D. polystachya*, respectively), revealed that the IGS regions *trnC*–*petN*, *trnL*–*rpl32*, *ndhD*–*ccsA*, and exon 3 of *clpP*, with high nucleotide diversity values (*π* > 8.50 × 10^−3^) (Figure 6), could serve as candidate DNA barcodes for species/cultivar identification in the *Enantiophyllum*. The availability of these highly divergent hotspots also provides valuable genetic information for future phylogenetic, population, genetic, and phylogeographic studies in this section.

### 3.3. Phylogenetic Relationships within D. alata and among Species of the Enantiophyllum Section

China is an important and possibly isolated domestication center of *D. alata* [12]. In this study, a total of four whole-plastome haplotypes were detected across all 44 accessions from eight provinces of mainland China and Taiwan Island (Table 1). The distribution of these four haplotypes was random, supporting the previous finding that there was no significant geographical differentiation within the *D. alata* in China [12]. Such genetic variations among these cultivars would be helpful in guiding the choice of suitable parents for hybridization breeding and the subsequent utilization of this economically important species [6]. Among the four haplotypes, Hap I predominated in most regions of China, accounting for 56.8% of Chinese accessions. The accessions of Hap I may have been established from those having Hap II (e.g., CDa017 from Hainan province) via a single-base substitution in the coding region of *ndhD*, which discriminated Hap I from Hap II (Table 1, Figure 7). The phylogenetic analysis suggested that two accessions carrying Hap IV (CDa020 from Hunan province and CDa044 from Zhejiang Province) formed a basal clade with respect to the entire clade of remaining accessions (Figure 7). This may reflect that Hap IV had a relatively ancient origin and long-term evolutionary history, which enabled its adaptation to different climatic conditions, and thus, played an important role in the breeding programs of *D. alata*. However, since a plastome-based tree effectively represents the history of a single locus [40,41], further rigorous examination with multiple single-copy nuclear genes or whole genome sequences is needed.

In addition, it is also noteworthy that although our phylogenetic analysis based on whole plastome sequences provided much more phylogenetic resolution than those with traditional molecular markers, such as amplified fragment length polymorphisms (AFLPs), and several plastid (e.g., *atpB*, *matK*, *rbcL*, and *trn*L–*trn*F) and nuclear (*Xdh*) loci [1,37,38,39,42], this study was conducted based on insufficient taxa sampling of the section *Enantiophyllum*. Thus, further phylogenomic studies with comprehensive taxon sampling are needed to establish a complete picture of the phylogenetic relationships within this section and identify the closet relatives of *D. alata*.

## 4. Materials and Methods

### 4.1. Plant Samples, DNA Extraction, and Genomic Data Acquisition

According to our field observation and previous phenotype- and SSR-based studies [12,16], a total of 44 accessions (Table 1) representing the major cultivars of *D. alata* in China were selected for whole genome sequencing. These accessions, with high phenotypic diversity, covered the main distribution area of this species in China, from west (Lijiang, Yunnan Province, geographic coordinates (gc): 26°43′ N, 100°15′ E) to east (Taizhou, Zhejiang Province, gc: 28°38′ N, 121°27′ E), and from south (Sanya, Hainan Province, gc: 18°24′ N, 109°45′ E) to north (Xuzhou, Jiangsu Province, gc: 34°39′ N, 116°35′ E). In addition, we also specifically selected accessions that have greatly contributed to the breeding and research of this species and accessions highly resistant to abiotic and biotic stress. Genomic DNA from each accession was extracted from fresh or silica gel-dried leaves using a DNAsecure Plant Kit (Tiangen Biotech, Beijing, China), according to the manufacturer’s protocol. The DNA concentration and integrity were measured on an Agilent 2100 BioAnalyzer (Agilent Technologies, Palo Alto, CA, USA), together with agarose gel electrophoresis. Paired-end libraries with an insert size of 350 bp were constructed and then sequenced on the BGISEQ-500 platform to generate raw sequences with a 150 bp read length. The library construction and sequencing were conducted at Wuhan Benagen Tech Solutions Company Limited, Wuhan, China.

Additionally, whole-genome sequencing datasets encompassing eight African accessions of *D. alata* (Table 1) were downloaded from the National Centre of Biotechnology Information (NCBI) Sequence Read Archive (SRA) database and were converted to the FASTQ format using the fastq-dump utility from the SRA Toolkit v.2.9.6 [43].

### 4.2. Plastome Assembly and Annotation

The raw reads in the FASTQ format were trimmed to remove the adapters and low-quality sequences using Trimmomatic v.0.36 [44], with the default parameters. The remaining clean reads were utilized for de novo assembly of complete plastome sequences using GetOrganelle v.1.7.6 [45], with the following parameters: -R 15-k 21,45,65,85,105-F embplant_pt. The connection and circularity of the assembly graphs from GetOrganelle were subsequently visually checked in Bandage v.0.9.0 [46]. The plastome annotations were conducted using Geneious Prime 2022.0.1 (https://www.geneious.com, accessed on 16 November 2021) by aligning each newly assembled plastome to previously published plastomes of *D. alata* (MG267382) and *D. japonica* (MT920319), as references, and transferring the reference annotations to these new plastomes. The resultant annotations were further manually checked and adjusted for the accuracy of the start and stop codons and exon/intron boundaries.

### 4.3. Plastome Polymorphism Analysis

All the 52 plastome sequences were aligned using the MAFFT v.7 plugin [47] as implemented in Geneious Prime 2022.0.1. The whole-plastome sequence alignment in the NEXUS format was then imported into DnaSP v.6.12.03 [48] to calculate the number of polymorphic sites (*E*), the number of haplotypes (*H*), haplotype diversity (*H*d), nucleotide diversity (*π*), and Tajima’s *D*. A total of four whole-plastome haplotypes were identified (see Results) and deposited in GenBank (accession numbers: OP787123–OP787126). The circular plastome maps of *D. alata* were drawn with the web-based software OrganellarGenomeDRAW (OGDRAW) v.1.3.1 [49].

### 4.4. Comparative Plastome Genomics within D. alata and among Closely Related Species

To evaluate the genomic similarity of the whole plastomes within *D. alata* and among members of the section *Enantiophyllum*, all four whole-plastome haplotypes (Haps I–IV) identified from the 52 *D*. *alata* accessions, together with another five previously published *Enantiophyllum* plastomes [i.e., *D. brevipetiolata* (OL638495), *D. cirrhosa* (ON584759), *D. glabra* (OL638497), *D. japonica* (MT920319), and *D. polystachya* (KY996494)—one individual per species], downloaded from the NCBI database, were compared. All these nine complete-plastome sequences were aligned using the global alignment program Shuffle-LAGAN [50], and visualized using the mVISTA browser [51], with the Hap I of *D. alata* as a reference. The four junctions between the two invert repeat (IRs) and large/small single copy (LSC/SSC) regions, termed J_LA_, J_SA_, J_SB_, and J_LB_, were further inspected and compared with the Repeat Finder plugin in Geneious Prime 2022.0.1 (https://www.geneious.com/plugins/repeat-finder/, accessed on 4 August 2019).

### 4.5. Characterization of Dispersed Repeats and SSRs

The online program, REPuter [52], was used to determine the number and size of dispersed repeats in *D. alata* and its closely related species, including forward (direct), reverse, complement, and palindromic repeats. The constraints for all the repeat sequences were set as follows: (i) a hamming distance of 3, and (ii) a minimum repeat size of 30 bp (i.e., 90% or greater sequence identity). In addition, the MISA-web application [53] was used to detect the SSRs across *D. alata* and another five *Enantiophyllum* species, with thresholds (minimum numbers) of 10, 5, 4, 3, 3, and 3 repeat units, respectively, for the mono-, di-, tri-, tetra-, penta-, and hexa-nucleotide SSRs.

### 4.6. Identification of Divergent Hotspots

To explore the highly divergent regions for DNA barcoding and population-based studies of *D. alata* as well as other *Enantiophyllum* species, four whole-plastome haplotype sequences (Haps I–IV) of *D. alata*, and one plastome sequence each for another five *Enantiophyllum* species (i.e., *D. brevipetiolata*, *D. cirrhosa*, *D. glabra*, *D. japonica*, and *D. polystachya*) were aligned with the MAFFT v.7 plugin [47] in Geneious Prime 2022.0.1 with the default settings. The protein-coding sequences (CDS), intergenic spacer regions (IGS), and introns and tRNAs with an aligned length over 200 bp and at least one mutation were subjected to nucleotide diversity (*π*) analysis using DnaSP v.6.12.03 [48].

### 4.7. Phylogenetic Analyses

The phylogenetic relationships within and among *D. alata* and its closely related species were conducted using two datasets: whole-plastome sequences and 79 shared protein-coding regions from all 52 accessions of *D. alata* (Table 1), and one accession each for another five *Enantiophyllum* species (i.e., *D. brevipetiolata*, *D. cirrhosa*, *D. glabra*, *D. japonica*, and *D. polystachya*), with *D. schimperiana* (MG805614) and *D. sansibarensis* (MG805614) as outgroups [39,54]. Both the whole-plastome sequences and protein-coding sequences were aligned using the MAFFT v.7 plugin [47] in Geneious Prime 2022.0.1. The best substitution model, GTR + I + G, for each dataset, determined by the Akaike Information Criterion (AIC) in jModelTest v.2.1.4 [55], was used for both maximum likelihood (ML) and Bayesian inference (BI) analyses. The ML analyses were performed using RAxML v.8.2.12, [56] available in the CIPRES Science Gateway v.3.3 (http://www.phylo.org/portal2/, accessed on 14 November 2010), with 1000 bootstrap replications. The BI analyses were performed using the program MrBayes v.3.2.7 [57], consisting of two independent runs of 1 × 10^6^ generations, with four independent Markov chain Monte Carlo (MCMC) chains each (i.e., one cold and three heated) and a sampling frequency of 1000 trees. The first 1000 trees were discarded as ‘burn-in’, and the remaining trees were used to construct a majority-rule consensus tree and estimate the posterior probabilities (PPs).

## Figures and Tables

**Figure 1 ijms-24-03341-f001:**
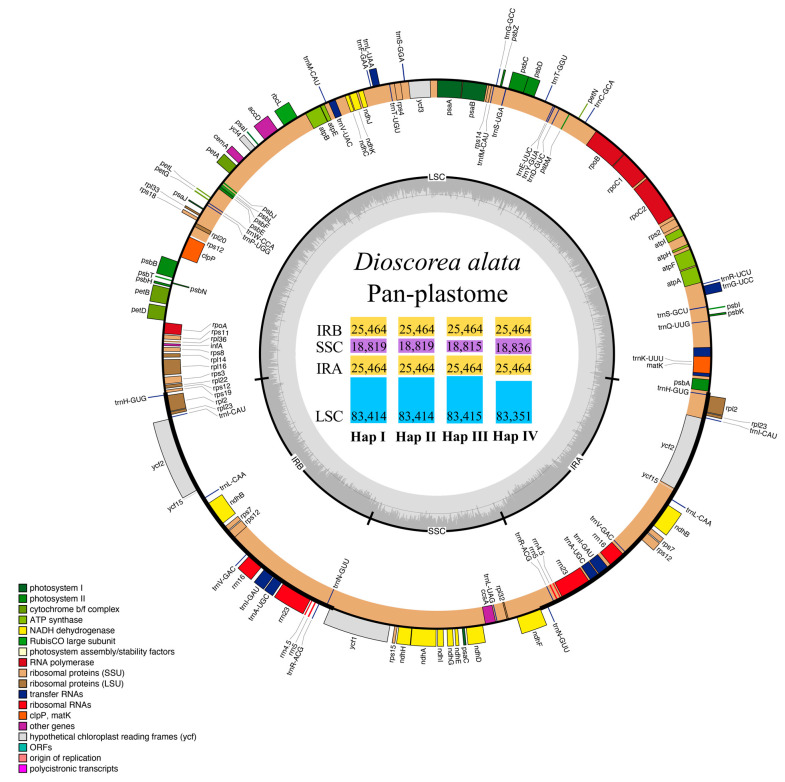
The pan-plastome map of *Dioscorea alata*. The genes shown on the outside of the circle are transcribed clockwise, while the genes inside are transcribed counter-clockwise. The genes belonging to different functional groups are coded by different colors. The GC/AT content is displayed by darker/lighter grey bars inside the tetrads (LSC, IRA, SSC, IRB). The lengths of LSC, IRA, SSC, and IRB regions (bp) of the four whole-plastome haplotypes (Haps I–IV) identified in *D. alata* were plotted in the inner circle.

**Figure 2 ijms-24-03341-f002:**
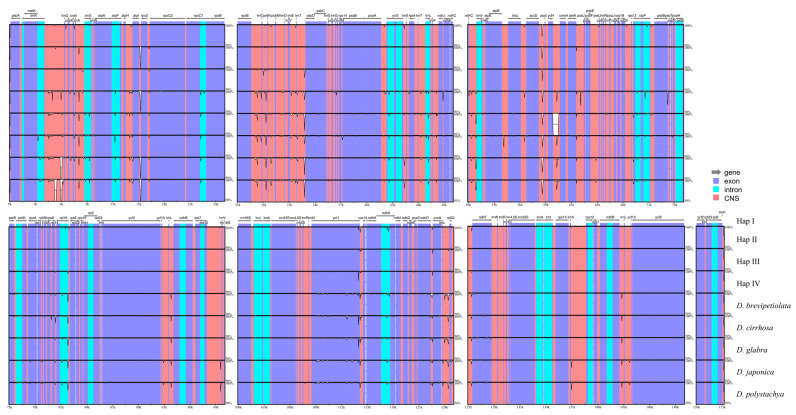
mVISTA-based identity plot showing plastome sequence identity within *Dioscorea alata* and among members of the section *Enantiophyllum*, with Hap I of *D. alata* as a reference. The vertical scale represents the percent identity between 50% and 100%. Annotated genes are displayed along the top, with grey arrows indicating their positions and transcriptional directions. Exons, introns, and conserved non-coding sequences (CNS) are highlighted in blue, cyan, and red, respectively.

**Figure 3 ijms-24-03341-f003:**
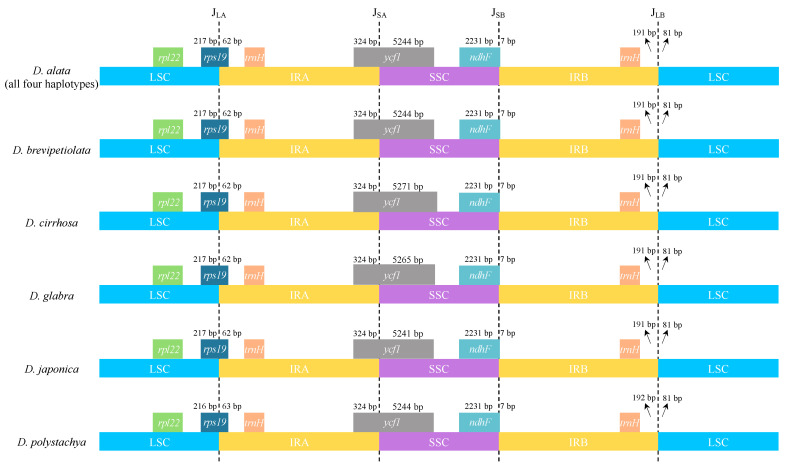
Comparison of IR/SC junctions among the nine *Enantiophyllum* plastomes (i.e., four whole-plastome haplotypes of *Dioscorea alata*, and one plastome each for *D. brevipetiolata*, *D. cirrhosa*, *D. glabra*, *D. japonica*, and *D. polystachya*, respectively). J_LA_, J_SA_, J_SB_, and J_LB_ refer to junctions of LSC/IRA, SSC/IRA, SSC/IRB, and LSC/IRB, respectively.

**Figure 4 ijms-24-03341-f004:**
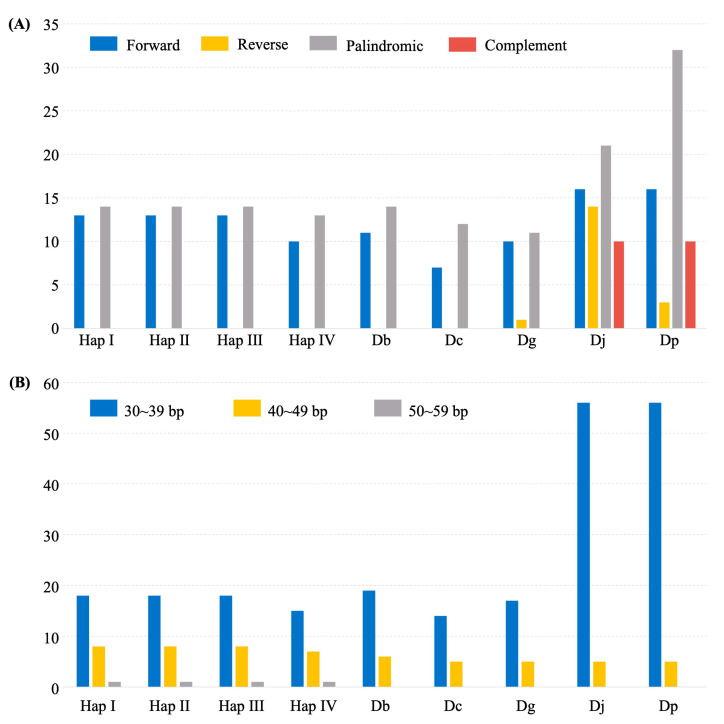
Analysis of dispersed repeats in the nine *Enantiophyllum* plastomes [i.e., four whole-plastome haplotypes of *Dioscorea alata*, and one plastome each for *D. brevipetiolata* (Db), *D. cirrhosa* (Dc), *D. glabra* (Dg), *D. japonica* (Dj), and *D. polystachya* (Dp), respectively]. (**A**) Numbers of four different repeat types; (**B**) frequency of dispersed repeats by length.

**Figure 5 ijms-24-03341-f005:**
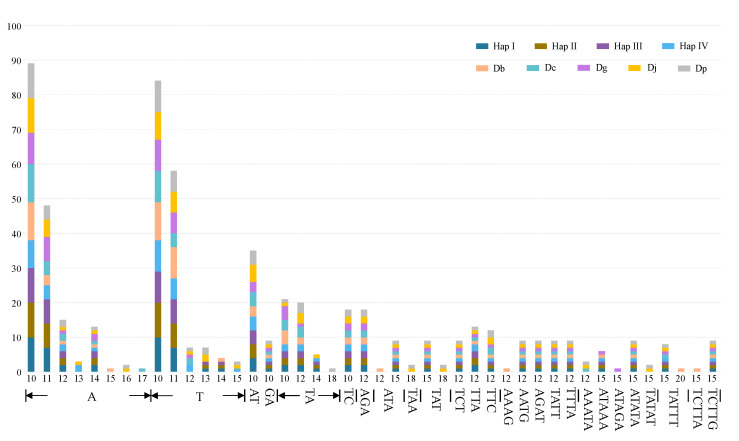
The motif types and lengths of SSRs in four whole-plastome haplotypes (Haps I–IV) of *Dioscorea alata* and another five *Enantiophyllum* plastomes. Abbreviations: Db, *D. brevipetiolata*; Dc, *D. cirrhosa*; Dg, *D. glabra*; Dj, *D. japonica*; Dp, *D. polystachya*.

**Figure 6 ijms-24-03341-f006:**
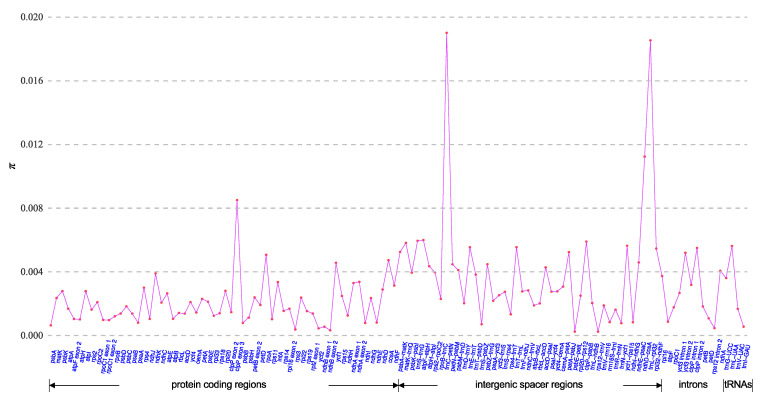
Nucleotide diversity (*π*) values across 120 regions (60 protein-coding regions, 45 intergenic spacers, 11 introns, and four tRNAs) extracted from the aligned plastomes of four whole-plastome haplotypes (Haps I–IV) of *Dioscorea alata* and another five *Enantiophyllum* plastomes.

**Figure 7 ijms-24-03341-f007:**
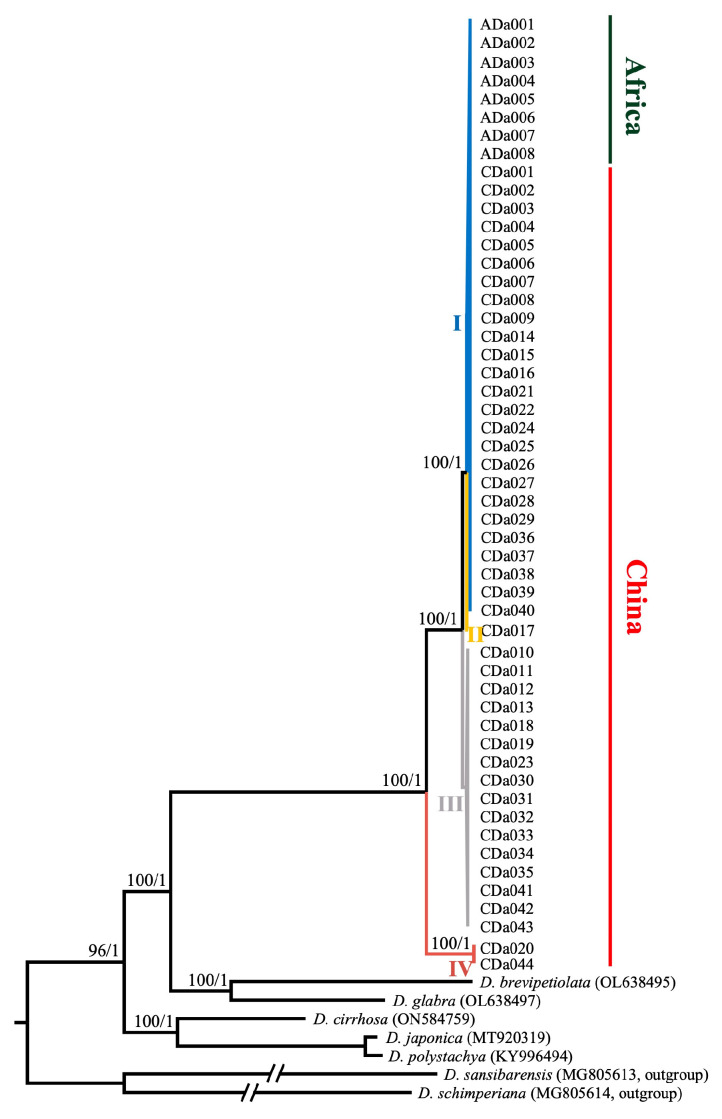
Phylogenetic relationships within *Dioscorea alata* and among members of *Enantiophyllum* inferred from maximum likelihood (ML) and Bayesian inference (BI) analyses based on whole-plastome sequences. Numbers at nodes represent ML bootstrap values (BS) and BI posterior probabilities (PP). The phylogenetic topologies resulting from two datasets (whole-plastome sequences and 79 shared protein-coding regions) are identical. All *D. alata* accessions could be divided into four clades (clades I–V), corresponding to the four haplotypes (Haps I–IV).

**Table 1 ijms-24-03341-t001:** Cultivar or local names, cultivation regions, and whole plastome haplotypes of *Dioscorea alata* accessions used in this study.

Accession No.	Cultivar or Local Name	Cultivation Region	Haplotype
CDa001	‘Minghuai No. 1’	Sanming, Fujian	I
CDa002	‘Quanhuai 1515’	Quanzhou, Fujian	I
CDa003	‘Quanhuai 1815’	Quanzhou, Fujian	I
CDa004	‘Shanghang dashu’	Fuzhou, Fujian	I
CDa005	‘Taihuai No. 6’	Quanzhou, Fujian	I
CDa006	‘Wuyishan zishanyao’	Wuyishan, Fujian	I
CDa007	‘Zhenghe hongxinshu’	Nanping, Fujian	I
CDa008	‘Zhouning zishu’	Ningde, Fujian	I
CDa009	‘Ziyushanyao’	Quanzhou, Fujian	I
CDa010	‘Anxi saobashu’	Quanzhou, Fujian	III
CDa011	‘Dongyou kuaishu’	Nanping, Fujian	III
CDa012	‘Minghuai No. 3′	Sanming, Fujian	III
CDa013	‘Zhenghe jiaobanshu’	Nanping, Fujian	III
CDa014	‘Ziyushenshu’	Guangzhou, Guangdong	I
CDa015	‘Hainan danshu’	Sanya, Hainan	I
CDa016	‘Hainan zazi’	Qiongzhong, Hainan	I
CDa017	‘Hainan zabai’	Wuzhishan, Hainan	II
CDa018	Da90	Lingao, Hainan	III
CDa019	Da94	Danzhou, Hainan	III
CDa020	‘Yangxi nuomishu’	Loudi, Hunan	IV
CDa021	‘Suyu No. 2’	Xuzhou, Jiangsu	I
CDa022	‘Suyu No. 4’	Xuzhou, Jiangsu	I
CDa023	‘Suyu No. 6’	Nanjing, Jiangsu	III
CDa024	‘Ganzhoushangyou baixinshu’	Ganzhou, Jiangxi	I
CDa025	‘Ganzhoushangyou zixinshu’	Ganzhou, Jiangxi	I
CDa026	‘Hongshuzi’	Ganzhou, Jiangxi	I
CDa027	‘Jiaobanshu’	Ganzhou, Jiangxi	I
CDa028	‘Wantian zishu’	Ruijin, Jiangxi	I
CDa029	‘Zixin yuantiao’	Shangrao, Jiangxi	I
CDa030	‘Baishu’	Ganzhou, Jiangxi	III
CDa031	‘Jiaobanshu purple’	Fuzhou, Jiangxi	III
CDa032	‘Jiaobanshu’	Ji’an, Jiangxi	III
CDa033	‘Longnante zipishanyao’	Ganzhou, Jiangxi	III
CDa034	‘Ximazhuang jiaobanshu’	Ruichang, Jiangxi	III
CDa035	‘Weihai baishu’	Weihai, Shandong	III
CDa036	‘Miyi shanyao’	Paizhihua, Sichuan	I
CDa037	‘Honglong shanyao’	Jiayi, Taiwan	I
CDa038	‘Yangmingshan shanyao’	Taibei, Taiwan	I
CDa039	‘Chengtuo baishanyao’	Wenshan, Yunnan	I
CDa040	‘Lijiang zishu’	Lijiang, Yunnan	I
CDa041	‘Quzhou kuaigenbaishu’	Quzhou, Zhejiang	III
CDa042	‘Ruian shanyao’	Wenzhou, Zhejiang	III
CDa043	‘Taizhou yuanshanyao’	Taizhou, Zhejiang	III
CDa044	‘Wencheng nuomishanyao’	Wenzhou, Zhejiang	IV
ADa001	Tda00/00005	Africa	I
ADa002	Tda01/00039	Africa	I
ADa003	Tda02/00012	Africa	I
ADa004	Tda05/00015	Africa	I
ADa005	Tda95-310	Africa	I
ADa006	Tda95/00328	Africa	I
ADa007	Tda99/00048	Africa	I
ADa008	Tda99/00240	Africa	I

Details of eight African accessions were reported in Bredeson et al. [27].

## Data Availability

The plastome data presented in this study can be found in the GenBank (https://www.ncbi.nlm.nih.gov/genbank/, accessed on 16 November 2021) under accession number OP787123–OP787126.

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
