# Peer review of "Pan-Plastome of Greater Yam (Dioscorea alata) in China: Intraspecific Genetic Variation, Comparative Genomics, and Phylogenetic Analyses"

_ijms, 2023, doi:10.3390/ijms24043341_

Round 1

Reviewer 1 Report

This study explored the intra-specific genetic variation with an important crop Dioscorea alata and its phylogenetic relationship with a few closely related species within the Section Enantiophyllum. The analysis confirmed that the samples collected from Africa could have originated from the ones cultivated in China. Also, phylogenies showed that D. alata is more closely related to D. brevipetiolata and D. glabra compared to D. cirrhosa, D. japonica and D. polystachya. The study is well-designed and the manuscript is relatively well-prepared. However, before it could be accepted for publication, some revisions will still be needed.

  First, it seems that more details on the origin of the samples need to be provided. Not only the names of the provinces should be given, but also the names of more accurate localities where they were collected. Ideally, the geographical coordinates should be provided.   Second, it would be better if the language of the entire manuscript could be checked, as some of the sentences seem to be not very clear. For example, L91-98.   Third, the font size for Figure 2 should be increased as at present they are almost invisible.    More specific issues: 1. Throughout the text, it seems that all "Enantiophyllum" should be italic, because it is Latin. L62: "method and principal", "method" may have to be deleted. L103: It seems that "including diverse Chinese and African cultivars/landraces. " could be deleted. L360-364: Because the information has been included in Table 1, there is no need to repeat it in the text. L465: "fromSanming", please insert a whitespace before "S".  

Reviewer 2 Report

This MS sets out to achieve 4 objectives connected to understanding infraspecific genetic variation in D. alata in China, its genomics and relationships to potential wild relatives via phylogenetics.

The infraspecific aspects of the MS appear to be scientifically valuable (e.g. the identification of potential barcodes, the 4 haplotypes) and well conducted methodologically.

I have significant reservations about phylogenetic component due to the limited breadth of sampling. The depth is much better, of course, than the studies referred to that were based on Sanger sequencing. I don’t think any significant conclusions can be drawn from the tree presented unfortunately.

Infraspecific sampling justification is required. Was landrace diversity adequately covered as indicated by e.g. vernacular name diversity or another metric?  The findings about D. alata diversity in China are only relevant and useful if landrace diversity has been adequately sampled and this needs to be evidenced.

The diversity of the African sampling is unknown but likely very limited when compared with the range of genotypes introduced across the continent.

Parts of the MS in particular the Discussion and perhaps also the Introduction could be shortened. The former especially where it is speculative or not well evidenced (e.g. phylogenetics, diversity of African accessions).

I have made multiple minor points and raised questions for the authors to address on the uploaded MS, including indicating some sections that should be reduced in length.

Reviewer 3 Report

It's not clear how pan-plastome was constructed in methods. It's usualy used tools as GET_HOMOLOGUES, and using tools as bidirectional blast hist, OrthoMCL, or COGtriangles.

why not use a control specie to compare this conservation and set the differences in this genera?

In figure 1 legend, it's not clear waht is referring the numbers above HapI, II, III, IV, in blue, purple and yellow.

It's not a strong support that only one SNP in an entire plastome is sufficient to think there is a different haplotype (Haplotype II), when only one accession has this change. The sequencing evidence for haplotype II (CDa017) is strong enough to support this SNP? Could it will be a sequencing error?

Could we talk about a polymorphic plastome instead different haplotypes?

Why the authors don't use variant calling tools against a reference plastome to determine polymorphism in al samples? (e.g. GATK).With variant calling tools we would know the coverage for all the accessions and to determine the significance of indels/SNP's (e.g. DepthOfCoverage tool)

Why don't the authors test the reported tools to identify haplotypes to have a parameter of significance for haplogroups? (e.g. HaplotypeCaller, VCFtools, SnpEff ...)

There is not much emphasis in the PAN part, there is the need to compare to previously reported studies, even in another species

Round 2

Reviewer 3 Report

I agree with the changes and answers from the authors.